# Cohort profile of the Sloane Project: methodology for a prospective UK cohort study of >15 000 women with screen-detected non-invasive breast neoplasia

Karen Clements ,[1] David Dodwell,[2] Bridget Hilton,[1] Isabella Stevens-Harris,[3] Sarah Pinder,[4,5] Matthew G Wallis ,[6,7] Anthony J Maxwell,[8,9] Olive Kearins,[1] Mark Sibbering,[3] Abeer M Shaaban,[10] Cliona Kirwan,[9,11] Nisha Sharma,[12] Hilary Stobart,[13] Joanne Dulson-Cox,[1] Janet Litherland,[14] Senthurun Mylvaganam,[15] Elena Provenzano,[7,16] Elinor Sawyer,[5] Alastair M Thompson[17]

For numbered affiliations see end of article.

**Correspondence to**
Karen Clements;
Karen.Clements@nhs.net

## ABSTRACT

**Purpose** The introduction of breast screening in the UK led to an increase in the detection of non-invasive breast neoplasia, predominantly ductal carcinoma in situ (DCIS), a non-obligatory precursor of invasive breast cancer. The Sloane Project, a UK prospective cohort study of screen-detected non-invasive breast neoplasia, commenced in 2003 to evaluate the radiological assessment, surgical management, pathology, adjuvant therapy and outcomes for non-invasive breast neoplasia. Long-term follow-up and accurate data collection are essential to examine the clinical impact. Here, we describe the establishment, development and analytical processes for this large UK cohort study.

**Participants** Women diagnosed with non-invasive breast neoplasia via the UK National Health Service Breast Screening Programme (NHSBSP) from 01 April 2003 are eligible, with a minimum age of 46 years. Diagnostic, therapeutic and follow-up data collected via proformas, complement date and cause of death from national data sources. Accrual for patients with DCIS ceased in 2012 but is ongoing for patients with epithelial atypia/in situ neoplasia, while follow-up for all continues long term.

**Findings to date** To date, patients within the Sloane cohort comprise one-third of those diagnosed with DCIS within the NHSBSP and are representative of UK practice. DCIS has a variable outcome and confirms the need for longer-term follow-up for screen-detected DCIS. However, the radiology and pathology features of DCIS can be used to inform patient management. We demonstrate validation of follow-up information collected from national datasets against traditional, manual methods.

**Future plans** Conclusions derived from the Sloane Project are generalisable to women in the UK with screen-detected DCIS. The follow-up methodology may be extended to other UK cohort studies and routine clinical follow-up. Data from English patients entered into the Sloane Project are available on request to researchers under data sharing agreement. Annual follow-up data collection will continue for a minimum of 20 years.

## STRENGTHS AND LIMITATIONS OF THIS STUDY

⇒ Large national prospective cohort study with high-quality clinical data and standardised reporting.
⇒ Long-term follow-up data collection allows for the detection of subsequent cancers and analysis of outcome data.
⇒ Data from the Sloane Project can be used by researchers to improve knowledge about ductal carcinoma in situ (DCIS) and atypia and help to address concerns about overdiagnosis and overtreatment.
⇒ Development of follow-up methodology using routinely collected data to identify further events could be generalised to other cohort studies, clinical trials and cancer sites.
⇒ Coding in other linked datasets may be inaccurate leading to overestimation of relative proportion of DCIS recurring as invasive disease compared with recurring as DCIS.
⇒ Participation was voluntary and 100% case capture was not achieved.

## INTRODUCTION

The UK National Health Service Breast Screening Programme (NHSBSP) was established in 1988, originally to screen women aged 50–64 years every 3 years using single-view mammography. The NHSBSP now comprises individual programmes in each of the four devolved nations which have developed since the inception of UK-wide breast screening. Following the extension of the eligible age range to 70 years in England in

2002, Scotland in 2003–2004, Wales in 2006 and Northern Ireland in 2009, the NHSBSP now routinely invites all women in the UK from the age of 50 years to their 71st birthday to attend breast screening every 3 years.[1] In some areas of the UK, women aged 47–49 and/or 71–73 years received invitations for screening as part of an age extension trial,[2] and some women received an invitation at the age of 46 years as they were in their 47th year. Women from their 71st birthday onwards can self-refer every 3 years. Two-view mammography was introduced in Northern Ireland in 1989, Wales in 2001,[3] England in 2003 and in Scotland in 2008/2009, and is now used in all centres.[4] Analogue mammograms were replaced by digital mammography from 2008, with 99% of UK NHSBSP units able to perform digital mammography as of October 2013.[5] Digitisation of NHSBSP mammography was completed in 2015.

The introduction of breast screening in the UK led to an increase in the detection of non-invasive breast neoplasia, predominantly ductal carcinoma in situ (DCIS).[6] A final diagnosis of DCIS now accounts for 20% of screen-detected breast cancer[7] and 11% of all breast cancers diagnosed in women aged 50 years and older.[8] A diagnosis of DCIS poses a difficult problem for patients and clinicians since there remains uncertainty about the natural history, rate of progression, invasive potential, optimal treatment and follow-up protocols, which has led to concerns about overdiagnosis and overtreatment.[6]

Some DCIS progresses to invasive breast cancer, but the risk varies with features of DCIS.[6 9]

Three randomised trials and one single-arm trial of active surveillance versus guideline-concordant surgery for 'low-risk' DCIS have been implemented, to investigate concerns about overdiagnosis and overtreatment.[10–13] In addition, an international study is using a wide range of approaches to address the issues of overdiagnosis and overtreatment as part of the Cancer Grand Challenge.[14]

An increased risk of both contemporaneous and subsequent invasive breast cancer is also seen in some other screen-detected lesions of uncertain malignant potential such as atypical intraductal epithelial proliferation/atypical ductal hyperplasia, flat epithelial atypia and lobular in situ neoplasia. However, prospective data on their management and outcomes are lacking, supporting the need for large, prospective studies, as data currently available are based on historical trial data[15–17] or retrospective studies.[18–21]

The Sloane Project, established in memory of Professor John Sloane, a breast pathologist,[22] commenced in 2003 as a UK-wide prospective cohort study of non-invasive breast neoplasia detected within the NHSBSP. The original aim was to accrue a cohort of 10 000 women with screen-detected DCIS and subsequently to inform optimal radiological assessment, surgical management, pathology handling and reporting (including the features of greatest clinical and prognostic importance) and adjuvant therapy.

The potential value of this cohort study has been recognised as providing evidence for clinical practice by the National Institute of Clinical Excellence, which recommended that patients with screen-detected DCIS be entered into the Sloane Project in the guideline 'Diagnosis and treatment of early breast cancer, including locally advanced disease' published in February 2009.[23] More recently, the Independent Breast Screening Review[6] recommended 'continued support for the Sloane Project which has an extensive database of screen-detected cases of DCIS and the long-term follow-up of these cases may well improve our understanding of this condition'.

This manuscript describes the Sloane Project process of case ascertainment, data cleaning and, specifically, methodology for verification and analysis of primary information and follow-up data. The latter elements, in particular, may be widely applicable for analyses of other large national datasets.

## COHORT DESCRIPTION

The first phase of the Sloane Project collected data on women diagnosed with screen-detected non-invasive breast neoplasia from 01 April 2003 to 31 March 2012. The ongoing second phase is continuing to collect information on patients with epithelial atypia/in situ neoplasia, but excluding DCIS, and commenced on 01 April 2012. The collection of follow-up and mortality data is an ongoing process and continues for all Sloane Project patients. The minimum age for study entry was 46 years, with no maximum age.

### Eligibility
Eligibility for entry into the Sloane Project is indicated in figure 1.

### Case ascertainment
The Sloane prospective cohort study is based on patient-level data collected by the NHS as part of the care and support of patients and is voluntarily submitted by UK NHSBSP units. Sloane Project cases were therefore matched to the data collected through the annual NHSBSP/Association of Breast Surgery (ABS) breast screening audit,[24] a national audit of all patients diagnosed through the NHSBSP, primarily designed to assess surgical performance with surgical Quality Assurance Standards, for the period 01 April 2003–31 March 2012, to understand the level of case ascertainment.

### Governance
The project is managed, and the data are collated, maintained and quality assured by the Screening Quality Assurance Service (SQAS), based in Birmingham, UK. It is overseen by a Project Steering Group (see the Acknowledgements section) comprising radiologists, pathologists, surgeons, managers, statisticians, oncologists and patient advocates.

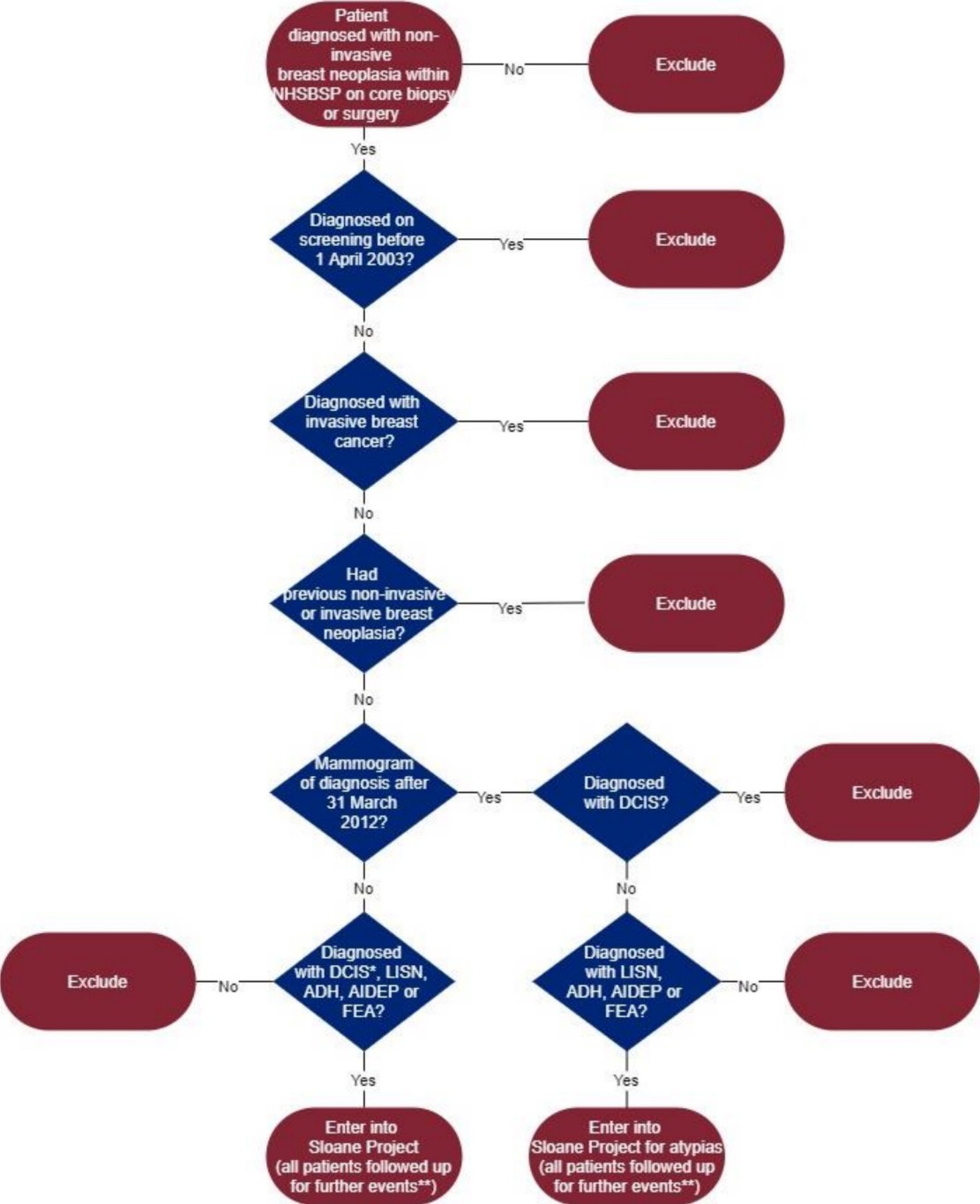

**Figure 1** Sloane eligibility flow chart. *Including DCIS with microinvasion (no more than 1 mm). **Further events include ipsilateral breast and nodal events, contralateral breast and nodal events and distant events, as well as deaths. ADH, atypical ductal hyperplasia; AIDEP, atypical intraductal epithelial proliferation; DCIS, ductal carcinoma in situ; FEA, flat epithelial atypia; LISN, lobular in situ neoplasia (including lobular carcinoma in situ and atypical lobular hyperplasia); NHSBSP, National Health Service Breast Screening Programme.

More recently, the study has been permitted to process personally identifiable data without consent under Regulation 5 of Statutory Instrument 2002 No. 1438: The Health Service (Control of Patient Information)[25 25] (15/CAG/0207) in line with the following clause: 'quality assuring screening services to ensure they are effective and safe, and that any incidents are investigated and managed appropriately'. This statutory exemption to common law permits the processing of personally identifiable data, as part of the core remit of population screening.

Data are securely held within the organisation with patient identifiers recorded. There is restricted access to a certified group of individuals and a full audit trail for access. A system-level security policy is in place to cover

the Sloane Project as a system asset as well as a data protection impact assessment. Figure 2 describes example of data flows in and out of the Sloane Project, along with the controls in place to mitigate for risks.

### Patient and public involvement

Patients from Independent Cancer Patients' Voice[26] joined the Sloane Project Steering Group in 2012. Since this time, they have actively contributed to publications guidance, communications and promotional material, including the website, as well as supporting funding applications. They are also involved, as part of a wider international collaboration, with the research that has followed from the cohort study.

Clinical follow-up data from the English programme are accessed via a data sharing agreement (DSA) between the National Disease Registration Service (NDRS) and SQAS. Follow-up and mortality data from Scotland, Wales and Northern Ireland need specific permissions. The agreements (from these 'devolved nations') do not allow for further onward sharing of data at present. Apart from the limitation described above, access to the Sloane Project data from external parties is governed by application to the breast screening Research Advisory Committee and Office for Data Release. Data will only be released by the Sloane Project to researchers under approval and in an anonymised or depersonalised format and under a DSA.

### Primary data capture

Demographic data and data deriving from radiology, pathology, surgery and adjuvant treatments at the time of diagnosis and initial treatment were originally captured via individual, pathology, surgery and radiotherapy paper proformas (see online supplemental file 1), collected at NHSBSP unit and hospital level. More recently, the proformas have been shortened and combined into one form, which can be completed electronically (see online supplemental file 2).[27] The forms are submitted to the Sloane Project team and data are then entered on a secure database held on an SQL server that generates an individual patient and tumour identifier.

As all Sloane patients are diagnosed within the NHSBSP, adherence to NHSBSP guidelines[28] and participation in the relevant quality assurance programmes[29] are mandatory. Participating units are required to follow a pathology protocol (see online supplemental files 3, 4)[30] that includes guidance on the handling and reporting of specimens, definitions for DCIS and atypia, microinvasion, cytonuclear grade, comedo necrosis and assessment of excision margins to NHSBSP pathology standards.[31] Radiology guidelines (see online supplemental files 3, 4)[32] encourage participating radiologists to complete the radiology proforma. Radiologists participate in the NHSBSP Personal Performance in Mammographic Screening external quality assurance scheme.[33]

### Data handling

Data are manually transcribed into the database, which has built-in integrity checks run on a regular basis, along with extra validation checks for each subsequent data request. Missing data items are subsequently added as far as possible. If the data cannot be sourced from information already supplied or from the NHSBSP unit or other data sources, then unknown/unrecorded status is assigned. Best efforts are made to minimise the number of missing data items (see online supplemental file 5).

Validation of data is undertaken by cross-checking with original NHSBSP unit source documents for those patients with recurrence and more generally, for the overall dataset, against the ABS national audits.[24] The Sloane Project data, as completed by clinicians, are taken as the primary source when there is conflicting information. However, the ABS national audit is used to supplement missing data items.

### Classification of subsequent events

Given the potential difficulties in distinguishing local recurrence from a new primary lesion in the same breast, terminology has been developed to classify subsequent events (online supplemental table 1).

### Death classification

Date and cause of death have historically been obtained directly from the NHS Trusts or Health Boards. Since 2013, the majority are now sourced from the NDRS/NHS Wales Cancer Network Information System Cymru data provided by the Office for National Statistics (ONS),[34] National Records of Scotland[35] and Northern Ireland patient records. Rules for coding the underlying cause of death are based on the rules used by ONS[34] that apply the condition or conditions entered in the lowest completed line of part I of the Medical Certificate Cause of Death. Women who die of breast cancer but who have no intervening breast events recorded are deemed to have had distant metastases on the date they died. The definitions for cause of death in online supplemental table 2 are used for the Sloane dataset.

### Follow-up and recurrence data capture

A methodology has been developed to identify further events, including ipsilateral breast/nodal, contralateral breast/nodal and distant events. This is run annually for the English patients, who comprise 87.1% of the Sloane cohort, and on an ad hoc basis for those from Scotland, Wales and Northern Ireland, who make up 10.1%, 2.5% and 0.3% of patients, respectively.

Further events are identified by matching women by their unique NHS number (in England and Wales) and Community Health Index number (in Scotland and Northern Ireland) and information provided by NHSBSP units to routinely collected UK datasets, as detailed in table 1.

The current version of the International Classification of Diseases and Related Health Problems[36] was used when

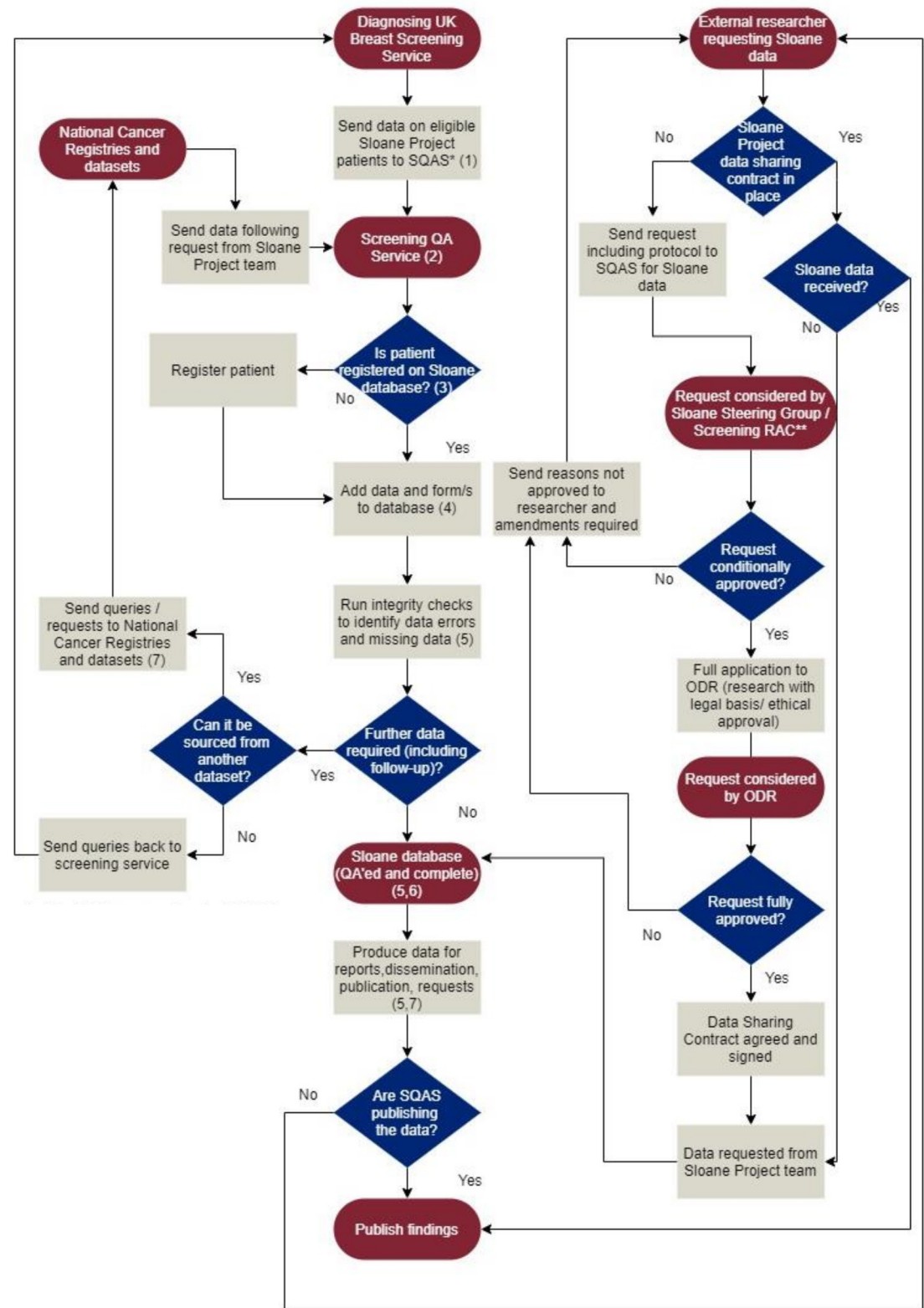

**Figure 2** Example of data flow diagram and key for Sloane Project data. *Screening Quality Assurance Service (SQAS). **Screening Research Advisory Committee (RAC). (1) Ensure follow SQAS guidelines for secure posting or email. (2) Use detailed internal protocols and policies for all aspects of Sloane work. (3) Avoid duplication by checking at least two identifiers. (4) Assign tumour and patient ID to minimise the use of personal data when working with the data. (5) Use system integrity checks and validation checks to indicate problems with data or quality assurance purposes. (6) System-level security policy in place, which includes audit trail, user access limits, server security among other things. Data protection impact assessment also in place. (7) With whom we have an existing data sharing contact. (8) Ensure publication is seen and approval given by Sloane Steering Group before publishing. Data release functions to change from 01 October 2021, but the process will remain similar, ODR, Office for Data Release.

**Table 1** National datasets provided to record follow-up and death information

| Country | Dataset used for Sloane follow-up data | Data collected | Data items used in Sloane English patient follow-up methodology |
|---|---|---|---|
| England (National Disease Registration Service incorporating the National Cancer Registration and Analysis Service)[57] | English Cancer Analysis System | Cancer registration data including diagnosis, treatment and tumour characteristics | dDate of diagnosis/treatment, trust/provider of diagnosis/treatment, ICD-10, morphology and behaviour, laterality, nodal positivity, adjuvant treatment, Office of Population Censuses and Surveys Classification of Interventions and Procedures (OPCS-4) |
| | Hospital Episode Statistics-admitted patient care[58] | Diagnosis and treatment | Date of admission/operation/episode start, ICD-10, OPCS-4, laterality, provider code |
| | Cancer Waiting Times[59] | Diagnosis and treatment | Date first seen/treatment start date, ICD-10, laterality, cancer treatment modality, cancer or symptomatic breast referral patient status, Cancer Treatment Event (CTE) type and metastatic site |
| | English National Radiotherapy dataset[60] (2009 onwards) | Adjuvant radiotherapy | Treatment/appointment start date, treatment region, ICD-10 |
| | Systemic Anti-Cancer Therapy dataset[61] | Adjuvant and neoadjuvant chemotherapy | Organisation code, ICD-10, morphology, drug treatment intent, start date, drug group |
| | Office for National Statistics mortality data[34] | Date and cause of death (diagnosed patients with cancer) | |
| | Mortality and Birth Information System | Date and cause of death (diagnosed atypia patients) | |
| Scotland–Information Services Division of the Scottish government and National Services Scotland[62] | Cancer registration data | Diagnosis, treatment, tumour characteristics, date of death and cause of death | |
| Wales–Cancer Network Information System[63] | Cancer registration data | Diagnosis, treatment, tumour characteristics, date of death and cause of death | |
| Northern Ireland | Sloane contact provides data from hospital records and systems on relatively small cohort of patients | Diagnosis, treatment, tumour characteristics, date of death and cause of death | |

ICD-10, International Classification of Diseases and Related Health Problems.

ascertaining diagnostic codes alongside Office of Population Censuses and Surveys Classification of Interventions and Procedures (OPCS-4)[37] when ascertaining procedure codes.

The censor date used in time-related analyses is the earliest of date of death, date of first further event (DCIS, invasive breast cancer, metastatic breast cancer) or end of time period of matched datasets if no further event has occurred and the woman is still alive.

**Follow-up methodology**

The follow-up methodology was developed to test the proof of concept of using the English National Cancer Registration and Analysis Service (NCRAS) data to identify further events and decrease the burden of local data collection within NHSBSP units and NHS Trusts.

The methodology is based on the following variables from the Sloane database:
► Date of mammogram.
► Date of surgery.
► Laterality of primary lesion.

These are then linked (via NHS number) to patients registered within the datasets and data items as described in table 1.

## Validation of the follow-up methodology

The methodology was validated on a subset of 3711 (39%) of the cohort of Sloane patients for whom recurrence information was independently collected directly from the NHSBSP units prior to the generation of the methodology using NCRAS data from 2013. These data were compared with NCRAS data focusing on invasive status (invasive/non-invasive/distant), laterality and date of diagnosis for the primary and any further event.

To confirm a match of either primary or recurrence, data were considered to be:

▶ *Acceptable*—same laterality, same invasive status, date of diagnosis within 6 months (either prior or after).
▶ *Not acceptable*—different laterality, different invasive status (with the exception of microinvasion which is coded as invasive on Cancer Analysis System (CAS)), dates >12 months apart.
▶ *Query acceptable*—for example, dates different between 6 and 12 months; for example, in one case the diagnosis date and NCRAS chemotherapy date (the only event) differed by 8 months, which was considered to be acceptable.

In order to support this proof-of-concept project and provide further validation, we sent a list of 100 Sloane Project patients to University Hospitals of Derby and Burton NHS Foundation Trust (UHDB). Some of these patients we had identified as having a further event using the methodology. The Trust was blinded to our records for each patient and was requested to identify those who they believed to be patients with a further event, and provide date and type, but also to indicate those who had not had a further event according to their records.

## FINDINGS TO DATE

To date, several papers[9 38–49] have been published on the Sloane cohort, which have shown that the radiology and pathology features of DCIS can be used to inform patient management. Outcome data also show that DCIS is not the indolent disease often described, and the need for longer-term follow-up for screen-detected DCIS has been confirmed.

### Case ascertainment

Overall, case ascertainment ranged from 0% to 93% by NHSBSP unit (median 32%, mean 38%). Individual attributes (including patient, tumour and treatment characteristics), such as age at screening (table 2), cytonuclear grade of DCIS, DCIS size, oestrogen receptor status, surgical and adjuvant treatment have been shown to be representative when comparing the voluntary Sloane prospective cohort study with the compulsory ABS/NHSBSP annual screening audit, although the more detailed Sloane data represent 38% of patients with DCIS diagnosed through the NHSBSP over the same time period.

### Primary data completeness

Data completeness, assessed against a pragmatic desired completeness level of ≥95%, demonstrated the majority

**Table 2** Age at screening mammogram for patients in the Sloane cohort compared with patients in the ABS cohort (p=0.81)

| Age at screening mammogram | Sloane cohort (%) | ABS/NHSBSP cohort (%) |
|---|---|---|
| <50 | 2.8 | 3.1 |
| 50–54 | 25.1 | 25.1 |
| 55–59 | 21.4 | 21.1 |
| 60–64 | 23.5 | 23.3 |
| 65–69 | 19.9 | 19.9 |
| 70+ | 7.3 | 7.5 |
| Total (known age at mammogram) | 100 | 100 |

ABS, Association of Breast Surgery; NHSBSP, National Health Service Breast Screening Programme.

of key data items for the patients with DCIS within the Sloane Project database have excellent completeness levels. Data completeness was high for most data fields (table 3).

### Follow-up methodology validation results

The results of the initial validation of the follow-up methodology, whereby recurrence (presence or absence of) information independently collected by Sloane was compared with routinely collected datasets from NCRAS, are shown in tables 4 and 5.

Table 4 shows that of 3711 Sloane patients with DCIS for whom follow-up data were received direct from the NHSBSP units, 91.5% were also in the NCRAS datasets and were an 'acceptable match'. There was a 96.4% match between the Sloane and NCRAS datasets on fact of 'no further event'. One hundred eight further events were identified through NCRAS datasets that had not been identified by the NHSBSP units. Further checking with the hospitals at which the event occurred confirms that these are correct. There are likely to be a number of reasons for this such as the patient's further event being diagnosed at a different NHS Trust or Health Board, or the type of hospital systems that were checked for further event and the likelihood of finding the information on those systems.

Table 5 shows more detail about those 18 recurrences not matched according to our rules by NCRAS datasets. Of these, 11 were identified in Hospital Episode Statistics (HES) but the invasive status was recorded incorrectly (9 of 11) or laterality was incorrect (2 of 11). Five were not identified in HES because two had surgery performed outside the NHS, one had no surgery due to comorbidity, two remained unknown, although may have had their surgery in the private sector. Two were not found on CAS or HES but were identified on the Cancer Waiting Times dataset, but invasive status was incorrect.

We concluded that although 92% events were successfully matched for further event correctly on NCRAS datasets, a further 13 recurrences (6%) would now be picked up by recent improvements in NCRAS and linkage to

**Table 3** Data completeness for key primary data items (patients with DCIS)

| Data item | | Data completeness of related field/s (%) of forms returned |
|---|---|---|
| Patient details | | |
| | Date of birth | 100.0 |
| | NHS/CHI number | 100.0 |
| | Hospital number | 96.0 |
| | Screening number | 99.6 |
| | Hospital of diagnosis or treatment | 98.2 |
| | Laterality of primary | 99.7 |
| Radiology | | 97.9 |
| | Site | 97.5 |
| | Background pattern | 97.5 |
| | Predominant radiological feature | 98.1 |
| | Presence/absence of microcalcification | 97.8 |
| | Type of calcification | 99.0 |
| | Radiological size | 98.1 |
| | Breast volume | 96.4 |
| | Specimen X-ray (cores of diagnostic surgical specimen) | 83.8 |
| Treatment | | 99.0 |
| | Recorded presence/absence of preoperative diagnosis of DCIS | 93.0 |
| | Surgical procedure (or no surgery confirmed) | 99.0 |
| | Date of surgical procedure (where relevant) | 99.8 |
| | Axillary procedure done/not done | 97.1 |
| | Axillary procedure type (where relevant) | 99.8 |
| | Adjuvant therapy (whether adjuvant therapy given or not) | 99.2 |
| Pathology | | 97.6 |
| | Core/vacuum-assisted biopsy | 81.6 |
| | Surgical specimen type | 96.2 |
| | Final histological diagnosis overall (from excision if excision done, from core if only core done) | 97.3 |
| | Cytonuclear grade on surgical specimen (DCIS) | 99.2 |
| | Architectural growth pattern (DCIS) | 91.7 |
| | Size (DCIS) | 98.4 |
| | Presence/absence of comedo necrosis (DCIS) | 89.7 |
| | Presence/absence of microinvasion (DCIS) | 97.4 |
| | Radial margin status (breast conserving surgery+DCIS) | 94.1 |
| | ER status (positive, negative or unknown (ER not done)) | 85.4 |
| | Progesterone (PR) status (included positive, negative or unknown (not done)) | 78.7 |
| | Human Epidermal Growth Factor Receptor 2 (HER2) status (included positive, negative or unknown (not done)) | 73.6 |
| Radiotherapy (yes/no) | | 85.3 |
| | Dose (where relevant) | 95.2 |
| | Energy (where relevant) | 91.8 |
| | Number of fractions (where relevant) | 94.5 |

CHI, Community Health Index; DCIS, ductal carcinoma in situ; ER, oestrogen receptor; NHS, National Health Service.

other datasets. It is likely that using currently available matching to national datasets, 206 of 211 patients (97%) would have accurate identification of DCIS or invasive breast cancer recurrence on the same side, contralateral breast or distant metastatic disease.

The further validation on 100 UHDB patients showed that using the Sloane follow-up methodology, 20 matched exactly for further events, 75 matched as having no further events and 5 did not match. Thus, 95% of patients were correctly matched. Out of the five that did not match, three were because the Sloane methodology had resulted in a 'query further event' and UHDB confirmed 'no further event', one was 'unknown invasive status'

and UHDB confirmed 'invasive' and one was recorded as 'ipsilateral DCIS' whereas UHDB had it recorded as 'ipsilateral invasive'. These results can be used to improve the methodology.

## STRENGTHS AND LIMITATIONS
### Strengths
Long-term follow-up of patients is required to ensure the impact of diagnosis, treatment and subsequent care is optimised and changes in management evaluated. Large, prospective cohort studies of conditions such as DCIS where subsequent events are relatively rare and occur

**Table 4** Validation of the follow-up methodology for identifying events/disease-free status (total number of patients=3711), comparing direct Sloane follow-up (211 patients with events identified) versus NCRAS follow-up (319 patients with events identified)

| NCRAS follow-up (using follow-up methodology) | Direct Sloane follow-up (n=3711) | |
|---|---|---|
| | No of patients identified with further event in Sloane (n=211) (% of events identified from Sloane) | No of patients without further event in Sloane (n=3500) (% of patients without further event from Sloane) |
| No of patients identified with further event in NCRAS (n=301) | 193 (91.5)[AM] | 108 (3.1)[NAM] |
| No of patients identified without further event in NCRAS (n=3377) | 4 (1.9)[NAM] | 3373 (96.4)[AM] |
| No of patients identified with further event different between the datasets (n=18) | 14 (6.6)[NAM] | 4 (0.1)[NAM] |
| No of patients not recorded on the NCRAS database (either primary or further event) (n=15) | 0 (0)[NAM] | 15 (0.4)[NAM] |

AM, acceptable match; NAM, not acceptable match; NCRAS, National Cancer Registration and Analysis Service.

over decades present particular challenges. Even for smaller cohorts and especially for clinical trials, the benefits of accurately matching prospectively collected data to national datasets may be invaluable. The ambitious, prospective, long-term Sloane Project has identified and followed some 15 000 patients with NHSBSP-detected pre-invasive breast neoplasia over almost two decades. We demonstrate here a high level of concordance between subsequent disease events confirmed in the Sloane cohort data (provided directly by NHSBSP units) and those identified using a follow-up methodology applied to electronically stored national datasets. While events were rare, representing only 2% of the total cohort, concordance between data sources was strong. There was also a 96% level of concordance for freedom from further events. This is encouraging but requires future validation, before the use of NCRAS data to determine freedom from further events can be regarded as definitive.

This methodology, if validated in other settings, and potentially further refined, may be generalisable to other cohort studies and for clinical trial follow-up. The long-term follow-up of patients beyond an initial 5 years is increasingly important in early invasive breast cancer, as well as DCIS, given the low frequency of events over an extended time period in these groups.

Although Sloane patients comprise one-third of those diagnosed with DCIS within the NHSBSP during the relevant time period, the approach laid out here confirms that the data quality and completeness are high and appear to be representative of the overall population of patients with screen-detected DCIS in the UK. Thus, conclusions derived from analyses of care and clinical outcomes from the Sloane Project[9 38–49] are robust and generalisable to all women in the UK with screen-detected DCIS. This is supported by a recently published study of over 35 000 women with screen-detected DCIS treated within the UK NHSBSP from its start until 2014.[50] Patient histopathology, treatment and outcomes reported reflect those curated within the more detailed Sloane cohort.

**Table 5** Number of Sloane patients identified with a recurrence reported directly from NHSBSP units that matched/did not match to the same recurrence in the NCRAS datasets (when applying the Sloane follow-up methodology)

| First Sloane further event | Number of Sloane patients identified with further event | Number matched to NCRAS datasets ('acceptable match') |
|---|---|---|
| Ipsilateral non-invasive | 84 | 70 (83%) |
| Ipsilateral invasive | 56 | 54 (96%) |
| Ipsilateral distant (with invasive/ non-invasive diagnosed at same time) | 2 | 2 (100%) |
| Distant | 6 | 5 (83%) |
| Contralateral non-invasive | 18 | 17 (94%) |
| Contralateral invasive | 42 | 42 (100%) |
| Contralateral invasive and distant (with invasive at same time) | 1 | 1 (100%) |
| Bilateral non-invasive | 2 | 2 (100%) |
| Total | 211 | 193 (91%) |

NCRAS, National Cancer Registration and Analysis Service; NHSBSP, National Health Service Breast Screening Programme.

### Limitations

Participation in data provision by NHSBSP units in the Sloane Project is voluntary, not specifically funded and 100% case capture was not achieved. However, comparison with the mandatory ABS/NHSBSP annual audits, which collect less detailed data, demonstrates that the Sloane dataset is representative of a cross-section of patients, geographical spread and of NHSBSP units' practice at the time of accrual.

Engagement has been challenged by lack of resources, particularly NHS staff time, available for long-term data collection. However, staff participants in the project receive Continuing Professional Development (CPD) credits and certificates for their participation in primary and follow-up data capture which is useful for annual appraisal and revalidation. The Sloane Project DCIS database remains open for completion of data for those cases with missing information (and is currently over 94% complete).

Additional challenges include different health service structures and processes across England, Scotland, Wales and Northern Ireland. Data regarding the small proportion of patients treated outside the NHS are limited. Furthermore, coding of DCIS in some datasets may be inaccurate, resulting in a potential overestimation of the relative proportion of DCIS that is recurring as invasive disease compared with that recurring as DCIS.

## COLLABORATION

The Sloane Project group welcomes applications from the UK, European Economic Area (EEA) and international organisations to collaborate and release data. However, any data releases are subject to a common governance framework (as summarised in figure 2), which will ensure the correct confidentiality provisions, legal permissions and ethical approvals are adhered to. At the time of writing, only Sloane data from the English NHS Breast Screening Programme are available to researchers.

## CONCLUSIONS

Overall, the Sloane prospective cohort study continues to evaluate changes in practice and outcomes for patients with DCIS as a result of publications of its own findings[48 49] as well as others' research[51 52] and national guidelines and recommendations, in order to ascertain the impact of such variations. Extending the value of this prospective cohort study further, the data collected have been used to develop and support innovative research proposals examining the underlying biological and imaging features of DCIS.[53 54] These peer-reviewed, grant-funded studies generated, for example, molecular data for comparison with the high-quality clinical data from the Sloane Project obtained through the methodology described here to investigate mechanisms behind the progression of DCIS to invasive breast cancer.[55] Thus, the NHSBSP prospective cohort study of non-invasive breast neoplasia, particularly DCIS (the Sloane Project),[27] continues to deliver insights into the clinical management of these conditions and resultant patient outcomes over time and can help address concerns about overdiagnosis and overtreatment.[10 56] Development of cross-validation of primary data, particularly patient outcomes, with the increasingly informative national datasets, points to the generalisability of this approach.

**Author affiliations**
[1]Screening Quality Assurance Service, NHS England, Birmingham, UK
[2]Nuffield Department of Population Health, University of Oxford, Oxford, UK
[3]Royal Derby Hospital, University Hospitals of Derby and Burton NHS Foundation Trust, Derby, UK
[4]Guy's Comprehensive Cancer Centre, Guy's & St Thomas' Hospitals NHS Foundation Trust, London, UK
[5]School of Cancer and Pharmaceutical Sciences, King's College London Faculty of Life Sciences and Medicine, London, UK
[6]Cambridge Breast Unit, Cambridge University Hospitals NHS Foundation Trust, Cambridge, UK
[7]NIHR Cambridge Biomedical Research Centre, Cambridge, UK
[8]Nightingale Centre, Manchester University NHS Foundation Trust, Manchester, UK
[9]NIHR Manchester Biomedical Research Centre, Manchester, UK
[10]Department of Histopathology, Queen Elizabeth Hospital Birmingham and University of Birmingham, Birmingham, UK
[11]Division of Cancer Sciences, The University of Manchester Faculty of Biology, Medicine and Health, Manchester, UK
[12]Breast Unit, St James's Hospital, Leeds Teaching Hospitals NHS Trust, Leeds, UK
[13]Independent Cancer Patients' Voice, London, UK
[14]West of Scotland Breast Screening Centre, Glasgow, UK
[15]New Cross Hospital, Royal Wolverhampton Hospitals NHS Trust, Wolverhampton, UK
[16]Department of Histopathology, Cambridge University Hospitals NHS Foundation Trust, Cambridge, UK
[17]Department of Surgical Oncology, Dan L Duncan Comprehensive Cancer Center, Baylor College of Medicine, Houston, Texas, USA

**Acknowledgements** This work uses patient data collected by the NHS as part of their care and support. The English data are collated, maintained and quality assured by the Screening Quality Assurance Service (SQAS) and follow-up data by the National Cancer Registration and Analysis Service. Sloane Project Steering Group: Professor Alastair Thompson (Chair of Sloane Steering Group)—Professor of Surgery, Baylor College of Medicine, Houston, Texas; Mrs Samantha Brace-McDonnell—Patient Advocate, Independent Cancer Patients' Voice; Mrs Karen Clements—Breast Cancer Research Manager, NHS England and NHS Improvement; Professor Sarah Darby—Professor of Medical Statistics, University of Oxford; Dr David Dodwell—Consultant Clinical Oncologist, University of Oxford, Oxford; Ms Joanne Dulson-Cox—National Audit Project Senior QA Officer, NHS England and NHS Improvement; Mrs Bridget Hilton—National Audit Project Senior QA Officer, NHS England and NHS Improvement; Mrs Olive Kearins—Quality Assurance Director, NHS England and NHS Improvement; Miss Cliona Kirwan—NIHR Clinician Scientist in Surgical Oncology and Consultant Oncoplastic Breast Surgeon, University Hospital of South Manchester; Dr Janet Litherland—Consultant Radiologist, West of Scotland Breast Screening Programme, Glasgow; Dr Anthony Maxwell—Consultant Radiologist, University Hospital of South Manchester, Manchester; Mr Seni Mylvaganam—Consultant Oncoplastic Breast Surgeon, Royal Wolverhampton Hospital; Professor Sarah Pinder—Professor of Breast Pathology, Guys and St Thomas' Hospitals, London; Dr Elena Provenzano—Lead Breast Histopathologist, Addenbrooke's Hospital, Cambridge; Professor Elinor Sawyer—Consultant Clinical Oncologist, Guys Hospital, London; Dr Abeer Shaaban—Consultant Pathologist, University Hospitals Birmingham; Dr Nisha Sharma—Lead Clinician Radiology, Leeds Teaching Hospital NHS Trust; Ms Hilary Stobart—Patient Advocate, Independent Cancer Patients' Voice; Dr Matthew Wallis—Consultant Radiologist, Addenbrooke's Hospital, Cambridge. We are also grateful for the help, support and professional advice provided by previous members of the Sloane Project Steering Group: Mr Hugh Bishop (previous Chair), Dr Julian Adlard, Miss Jessamy Bagenal, Mr Robert Carpenter, Professor John Dewar, Professor Ian Ellis, Professor Adele Francis, Professor W D George, Professor Andrew Hanby, Professor Sunil Lakhani, Dr Gill Lawrence, Mr Martin Lee, Dr James Macartney, Mr Stewart Nicholson, Professor Julietta Patnick, Dr Gillian Ross, Mr Mark Sibbering, Professor Valerie Speirs, Ms Sheila Stallard, Dr Jeremy Thomas, Professor Ian Tomlinson, Mrs Margot Wheaton, Ms Maggie Wilcox. All participating UK NHSBSP units: Avon, Aylesbury and Wycombe, Barnsley, Bedfordshire and Hertfordshire, Bolton, Breast Test Wales–North, Breast Test Wales–South East, Breast Test Wales–South West, Brighton, Cambridge and Huntingdon, Central & East London, Chelmsford and Colchester, Chester, City, Sandwell and Walsall, Cornwall, Crewe Derby City and South Derbyshire, Doncaster, Dorset, Dudley and Wolverhampton, East Berkshire, East Cheshire

& Stockport, East Lancashire, East Scotland, Gateshead, Gloucestershire, Great Yarmouth and Waveney, Greater Manchester, Guildford, Hereford and Worcester, Humberside, Isle of Wight, King's Lynn, Leeds and Wakefield, Leicestershire, Liverpool, Medway, Milton Keynes, Newcastle upon Tyne, Norfolk and Norwich, North and Eastern Devon, North and Mid Hampshire, North Derbyshire & Chesterfield, North East Scotland, North Lancs & South Cumbria, North London, North Midlands, North Nottingham, North Yorkshire, Northampton, Nottingham, Outer North East, London, Oxford, Pennine, Peterborough, Portsmouth, Rotherham Sheffield, Shropshire, Somerset, South Birmingham, South East London, South East Scotland, South Essex, South West London, South West Scotland, Southampton and Salisbury, Surrey (Jarvis), Warrington, Warwick, Solihull and Coventry, West Berkshire, West Devon, West Essex, West of London, West of Scotland, West Suffolk, Western BSU, Altnagelvin Hospital, Northern Ireland, Wiltshire, Wirral, Worthing (West Sussex), Public Health Scotland (previously the Information Services Division Scotland), NHS Wales Cancer Network Information System Cymru (CaNISC), Independent Cancer Patients' Voice.

**Contributors** KC (corresponding author and author responsible for the overall content as guarantor)—substantial contribution to the conception and design of the work, the acquisition, analysis and interpretation of data for the work; drafted and revised the work critically. DD—substantial contribution to the conception and design of the work, the acquisition, analysis and interpretation of data for the work; drafted and revised the work critically. BH—substantial contribution to the conception and design of the work, the acquisition, analysis and interpretation of data for the work; drafted and revised the work critically. IS-H—substantial contribution to the acquisition, analysis and interpretation of data for the work; revised the work critically. SP—substantial contribution to the conception and design of the work, the acquisition, analysis and interpretation of data for the work; revised the work critically. MGW—substantial contribution to the conception and design of the work, the acquisition, analysis and interpretation of data for the work; revised the work critically. AJM—substantial contribution to the conception and design of the work, the acquisition, analysis and interpretation of data for the work; revised the work critically. OK—substantial contribution to the conception and design of the work, the acquisition, analysis and interpretation of data for the work; revised the work critically. MS—substantial contribution to the acquisition, analysis and interpretation of data for the work; revised the work critically. AMS—substantial contribution to the acquisition, analysis and interpretation of data for the work; revised the work critically. CK—substantial contribution to the acquisition, analysis and interpretation of data for the work; revised the work critically. NS—substantial contribution to the acquisition, analysis and interpretation of data for the work; revised the work critically. HS—substantial contribution to the acquisition, analysis and interpretation of data for the work; revised the work critically. JD-C—substantial contribution to the acquisition, analysis and interpretation of data for the work; revised the work critically. JL—substantial contribution to the acquisition, analysis and interpretation of data for the work; revised the work critically. SM—substantial contribution to the acquisition, analysis and interpretation of data for the work; revised the work critically. EP—substantial contribution to the acquisition, analysis and interpretation of data for the work; revised the work critically. ES—substantial contribution to the acquisition, analysis and interpretation of data for the work; revised the work critically. AMT—substantial contribution to the conception and design of the work, the acquisition, analysis and interpretation of data for the work; drafted and revised the work critically. All authors listed in the manuscript have given final approval of the version to be published and agree to be accountable for all aspects of the work in ensuring that questions related to the accuracy or integrity of any part of the work are appropriately investigated and resolved.

**Funding** At the time of writing, the Sloane Project work is undertaken within the Screening QA Service at NHS England (no award/grant number). This work was supported by Cancer Research UK and by KWF Kankerbestrijding (ref.C38317/A24043) (KC,SP,HS,ES,AMT,MGW). The Sloane Project has received previous funding from the following: NHS Breast Screening Programme via Public Health England April 2013–September 2021, previously NHS (no award/grant number); Breast Cancer Research Trust—2-year project grant 2008–2009 (no award/grant number); Pfizer 3-year unrestricted educational grant 2005–2007 (no award/grant number); ad hoc fundraising events and private donations including Golf Charity Days (organised by Ken Ward), pub quizzes and fundraising balls (2005–2012) (no award/grant number).The work was supported by the NIHR Manchester Biomedical Research Centre (ref. IS-BRC-1215-20,007) (AJM and CK) and the NIHR Cambridge Biomedical Research Centre (ref. IS-BRC-1215-20,014) (MW and EP). The views expressed are those of the authors and not necessarily those of the NIHR or the Department of Health and Social Care.

**Disclaimer** The funders of the Sloane Project have not had any involvement in the study design; in the collection, analysis and interpretation of data; in the writing of the report; or in the decision to submit the article for publication.

**Competing interests** KC was funded (May 2019 to October 2022) as part of the Cancer Research Grand Challenge PRECISION team (C38317/A24043), which is funded by Cancer Research UK and the Dutch Cancer Society. DD is funded by Cancer Research UK (grant C8225/A21133). EP received speaker's honoraria (Roche), travel costs to speak at meeting (Roche), and participation in advisory group meetings for IPB advisors and Roche. SP participated in advisory boards for AstraZeneca, Roche and Exact Sciences, received money for speaking at meetings for Roche and Exact Sciences and is a member of the PRECISION consortium (C38317/A24043). AMS has participated in advisory boards for Exact Sciences and Veracyte. HS received travel and support to attend meetings of the CRUK Grand Challenge PRECISION Study (C38317/A24043). ES, AMT and MGW are members of the PRECISION consortium (C38317/A24043). BH, IS-H, AJM, OK, MS, CK, NS, JD-C, JL and SM have nothing to declare.

**Patient and public involvement** Patients and/or the public were involved in the design, or conduct, or reporting, or dissemination plans of this research. Refer to the Methods section for further details.

**Patient consent for publication** Not required.

**Ethics approval** Ethics Committee approval was assessed as not required for this study which was originally conducted under the NHS Cancer Screening Programme's application to the Patient Information Advisory Group (PIAG).

**Provenance and peer review** Not commissioned; externally peer reviewed.

**Data availability statement** Data are available upon reasonable request. Clinical follow-up data from the English Programme are accessed via a Data Sharing Agreement (DSA) between the National Disease Registration Service (NDRS) and SQAS. Follow-up and mortality data from Scotland, Wales and Northern Ireland need specific permissions. The agreements (from these 'devolved nations') do not allow for further onward sharing of data at present. Apart from the limitation described above, access to the Sloane Project data from external parties is governed by application to the breast screening Research Advisory Committee (RAC) and Office for Data Release (ODR). Data will only be released by the Sloane Project to researchers under approval and in an anonymised or depersonalised format and under a data sharing agreement.

**ORCID iDs**
Karen Clements http://orcid.org/0000-0003-0113-4409
Matthew G Wallis http://orcid.org/0000-0001-7141-281X

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
