## [Reviewer comments · BMJ Open]

ARTICLE DETAILS

TITLE (PROVISIONAL)	Cohort profile of the Sloane Project – methodology for a prospective UK cohort study of >15,000 women with screen-detected non-invasive breast neoplasia
AUTHORS	Clements, Karen; Dodwell, David; Hilton, Bridget; Stevens-Harris, Isabella; Pinder, Sarah; Wallis, M. G.; Maxwell, AJ; Kearins, Olive; Sibbering, Mark; Shaaban, Abeer; Kirwan, Cliona; Sharma, Nisha; Stobart, Hilary; Dulson-Cox, Joanne; Litherland, Janet; Mylvaganam, Senthurun; Provenzano, Elena; Sawyer, Elinor; Thompson, Alastair M

VERSION 1 – REVIEW

REVIEWER	Karsten Juhl Jørgensen Rigshospitalet, The Nordic Cochrane Centre
REVIEW RETURNED	21-Jul-2022

GENERAL COMMENTS	This manuscript describes the Sloane Project and the data material it collects. This projects has been the basis for a number of research projects intended to explore the natural history of DCIS and other non-invasive lesions detected through the UK mammography screening programmes. The importance of this project lies in the recognition that many (about 20%) of the lesions identified through breast screening likely needs to be managed differently than invasive cancers identified through the programme. Some subgroups of these non-invasive may not need treatment as they will never develop into conditions that cause morbidity or mortality. Indeed, several randomised trials, such as the LORIS and and WISDOM trials, are currently ongoing to explore strategies of monitoring the lesions rather than treating them. These trials and their relevance to the objectives of the Sloane Projects are currently not mentioned in the manuscript but in my opinion should be. An important observation is that although 20% of breast lesions detected through screening are DCIS lesions, the vast majority of which has traditionally been treated with surgery etc., we have not seen a reduction in the incidence of invasive breast cancers, although the programme has run for more than 30 years. This is in stark contrast to observations from the cervical cancer screening programme where the removal of non-invasive precursor lesions have led to very substantial reductions in the incidence of invasive cervical cancers. This is a hallmark of overdiagnosis, the most important harm of breast screening. To my mind, the most important objective of the Sloan Project would be to contribute to our knowledge about DCIS in a way that would allow us to reduce this critical problem with breast screening. That a substantial proportion of DCIS is currently overdiagnosed and overtreated is now generally accepted and the reason that trials like LORIS and WISDOM are
--

	necessary and ethically approved. It therefore seems curious to me that overdiagnosis with breast screening and DCIS is not mentioned at all or that the potential for the Sloan Project to inform about it regarding DCIS is not discussed. Apart from these omissions, the manuscript is very well written, clearly laying out the organisational details, data sources, and limitations of the Sloan Project. Karsten Juhl Jørgensen
--	--

REVIEWER	Seema A. Khan Northwestern University
REVIEW RETURNED	04-Sep-2022

GENERAL COMMENTS	This manuscript from the investigators involved in the Sloane Project is intended to describe the methods of case ascertainment, data collection and cleaning, and verification of follow-up and outcomes for the completed cohort of DCIS patients, and the ongoing cohort of individuals with atypical lesions of the breast. The DCIS cohort was completed in 2012, and several reports on cohort characteristics and outcomes have already been published. Overall, the Sloane project is an important enterprise and will continue to provide useful information regarding the invasive breast cancer potential of a diagnosis of DCIS and atypical lesions of the breast. A description of data collection and follow-up/outcome verification will be useful for the interpretation of data contained in future publications. The abstract does not follow the structure prescribed in the instructions for authors. Data sharing agreements from the ‘devolved nations’ do not allow for further onward sharing of data at present. Therefore, it appears that follow-up data will only be available from cohort participants in England. If this is true, it should be specifically stated in this and other publications. If not true, please clarify. The proformas presented in the supplementary materials are extremely detailed. They require significant commitment of data abstraction time. How is this supported? “As all Sloane patients are diagnosed within the NHSBSP, adherence to NHSBSP guidelines and participation in the relevant quality assurance programmes are mandatory”. Mandatory adherence to quality guidelines is an admirable requirement. But it seems likely that there would be instances of lack of adherence, many of which may be justified based on individual patient characteristics and preferences. It would be useful to know what level of nonadherence to guidelines results in exclusion of data. On reading prior Sloane project publications, it is not clear that quality-based exclusions were specifically defined. But if there are standards used for data exclusion, some explanation of those seems reasonable in this manuscript, which seems to be intended as a guide to future publications. The overall case ascertainment is 32%. The cohort is described as being representative, but no demographic details are provided.... Age? Race/ethnicity? Socioeconomic status? Family history?
--

	The use of NCRAS data for follow-up was instituted in 2013, using 3711 cohort participants from England. If I'm reading Table 2 correctly, it suggests that 211/3711 patients were identified as experiencing further events in Sloane, and an additional 108 were identified in NCRAS, but these were called "not acceptable match". That would mean that one-third of subsequent events in the Sloane cohort are missed. The possible reasons for this are described (subsequent care at a different NHS Trust or Health Board, the likelihood of finding the information in other hospital systems). Is it possible to verify that the 108 participants identified in NCRAS but not in Sloane did in fact experience subsequent events? This seems necessary for the use of NCRAS data going forward. r The further validation of Sloane data on 100 individuals recruited from University Hospitals of Derby and Burton NHS Foundation Trust (UHDB) yielded good data concordance, but some estimation of the statistical confidence in this estimate would be helpful. In the event definition table: What does contralateral re/occurrence mean? If contralateral recurrence is included in this term, it implies that a prior contralateral DCIS or invasive cancer had been diagnosed. But from the inclusion criteria it would appear that someone with prior contralateral disease would've been excluded. So in fact, according to the cohort entry criteria, only a new contralateral event would be relevant. In the same table, circulatory non-circulatory deaths are distinguished. What is the purpose of this distinction? For non-UK practitioners, some terms will need explanation. For instance, it is not clear what is meant by a diagnostic versus a therapeutic specimen radiograph.
--	---

VERSION 1 – AUTHOR RESPONSE

Reviewer: 1

Dr. Karsten Juhl Jørgensen, Rigshospitalet Comments to the Author:

This manuscript describes the Sloane Project and the data material it collects. This project has been the basis for a number of research projects intended to explore the natural history of DCIS and other non-invasive lesions detected through the UK mammography screening programmes.

The importance of this project lies in the recognition that many (about 20%) of the lesions identified through breast screening likely needs to be managed differently than invasive cancers identified through the programme. Some subgroups of these non-invasive may not need treatment as they will never develop into conditions that cause morbidity or mortality. Indeed, several randomised trials, such as the LORIS and WISDOM trials, are currently ongoing to explore strategies of monitoring the lesions rather than treating them. These trials and their relevance to the objectives of the Sloane Project are currently not mentioned in the manuscript but in my opinion should be. An important observation is that although 20% of breast lesions detected through screening are DCIS lesions, the vast majority of which has traditionally been treated with surgery etc., we have not seen a reduction in the incidence of invasive breast cancers, although the programme has run for more than 30 years. This is in stark contrast to observations from the cervical cancer screening programme where the removal of non-invasive precursor lesions have led to very substantial reductions in the incidence of invasive cervical cancers. This is a hallmark of overdiagnosis, the most important harm of breast screening. To my mind, the most important objective of the Sloan Project would be to contribute to our knowledge about DCIS in a way that would allow us to reduce this critical problem with breast screening. That a substantial proportion of DCIS is currently overdiagnosed and overtreated is now generally accepted and the reason that trials like LORIS and WISDOM are necessary and ethically approved. It therefore seems curious to me that overdiagnosis with breast screening and DCIS is not

mentioned at all or that the potential for the Sloan Project to inform about it regarding DCIS is not discussed.

We have added further text and references relating to overdiagnosis, overtreatment and the ongoing trials in the Article Summary – Strengths and Limitations, Introduction, and Conclusions.

Apart from these omissions, the manuscript is very well written, clearly laying out the organisational details, data sources, and limitations of the Sloan Project.

Karsten Juhl Jørgensen

Reviewer: 2

Seema A. Khan , Northwestern University Comments to the Author:

This manuscript from the investigators involved in the Sloane Project is intended to describe the methods of case ascertainment, data collection and cleaning, and verification of follow-up and outcomes for the completed cohort of DCIS patients, and the ongoing cohort of individuals with atypical lesions of the breast. The DCIS cohort was completed in 2012, and several reports on cohort characteristics and outcomes have already been published. Overall, the Sloane project is an important enterprise and will continue to provide useful information regarding the invasive breast cancer potential of a diagnosis of DCIS and atypical lesions of the breast. A description of data collection and follow-up/outcome verification will be useful for the interpretation of data contained in future publications.

The abstract does not follow the structure prescribed in the instructions for authors.

Thank you - the future plans section has been updated to include how the cohort will be used in the future along with planned follow up.

Data sharing agreements from the 'devolved nations' do not allow for further onward sharing of data at present. Therefore, it appears that follow-up data will only be available from cohort participants in England. If this is true, it should be specifically stated in this and other publications. If not true, please clarify.

Referee 2 is correct to point this out. It is specifically covered in Patient and Public Involvement section Paragraph 2. Abstract, article summary and Collaboration have been updated to make this clearer.

The proformas presented in the supplementary materials are extremely detailed. They require significant commitment of data abstraction time. How is this supported?

Primary data has been collected and entered manually by colleagues in breast screening units, sent to the Sloane Project team and then manually entered. The supplementary materials included forms that were used to capture data from the earlier part of the Sloane Project. The current proforma, which we have now added to the supplementary materials, capture all of the radiology, pathology and treatment data on one form (for atypia patients only as that is now the focus of new accruals), which lessens the burden for both clinicians and the Sloane Project team. We are currently piloting to the possibility of pre-completing some of the fields from other data sources prior to sending them to screening units.

The majority of follow-up data for DCIS are obtained from the National Cancer Registration and Analysis Service (NCRAS) data sources. Again, the recurrence form can be pre-completed with some data items prior to sending out to screening units to complete.

“As all Sloane patients are diagnosed within the NHSBSP, adherence to NHSBSP guidelines and participation in the relevant quality assurance programmes are mandatory”. Mandatory adherence to quality guidelines is an admirable requirement. But it seems likely that there would be instances of lack of adherence, many of which may be justified based on individual patient characteristics and preferences. It would be useful to know what level of nonadherence to guidelines results in exclusion of data. On reading prior Sloane project publications, it is not clear that quality-based exclusions were specifically defined. But if there are standards used for data exclusion, some explanation of those seems reasonable in this manuscript, which seems to be intended as a guide to future publications.

The NHSBSP guidelines relate to imaging and pathology diagnostic criteria and standards and are an annual requirement for individual clinician and screening unit appraisal and continued practice. These include imaging techniques and performance and a quality assurance slide set for breast screening pathology. There is inevitably some lack of adherence to guidelines, and this is documented at local and national level by QA teams and, where appropriate, remedial actions are taken.

Treatment by surgery, endocrine therapy and radiation therapy are driven by different national guidelines and clearly that is where patient preference and clinical judgement (eg administration of radiation therapy or prescription of endocrine therapy) become patterns of delivery of care. Variation in practice is documented annually by the NHSBSP and ABS audit of screen detected breast cancers (Audit - Association of Breast Surgery).

No data or patients are excluded as a result of 'non adherence' in fact many lessons can be learnt from variations in practice. We have previously published on variation in radiotherapy across the UK (Dodwell D, Clements K, Lawrence G, et al. Radiotherapy following breast-conserving surgery for screen-detected ductal carcinoma in situ (DCIS): Indications and utilisation in the UK: Findings from the Sloane Project. British Journal of Cancer 2007; 97: 725-729e) and are currently analysing further with an aim to publish in due course.

The overall case ascertainment is 32%. The cohort is described as being representative, but no demographic details are provided.... Age? Race/ethnicity? Socioeconomic status? Family history?

The Sloane cohort is representative of DCIS diagnosed in the UK in comparison with the Association of Breast Surgery breast screening audit of all patients diagnosed in the UK over the time frame of the Sloane prospective cohort study. To demonstrate this, the table below has now been incorporated into the paper for the future readership.

Age at screening mammogram	Sloane cohort (%)	ABS/NHSBSP cohort (%)
<50	2.8	3.1
50 – 54	25.1	25.1
55 – 59	21.4	21.1
60 – 64	23.5	23.3
65 – 69	19.9	19.9
70+	7.3	7.5
Total (known age at mammogram)	100	100

P=0.81

Unfortunately, data on race/ethnicity, socioeconomic status and family history has not been routinely collected by the NHSBSP. Ethnicity is now routinely collected, and this is something that we will be looking at in future work, as we become able to match to national datasets held by NCRAS for race/ethnicity, socioeconomic status for some of our Sloane patients.

The use of NCRAS data for follow-up was instituted in 2013, using 3711 cohort participants from England. If I'm reading Table 2 correctly, it suggests that 211/3711 patients were identified as experiencing further events in Sloane, and an additional 108 were identified in NCRAS, but these

were called “not acceptable match”. That would mean that one-third of subsequent events in the Sloane cohort are missed. The possible reasons for this are described (subsequent care at a different NHS Trust or Health Board, the likelihood of finding the information in other hospital systems). Is it possible to verify that the 108 participants identified in NCRAS but not in Sloane did in fact experience subsequent events? This seems necessary for the use of NCRAS data going forward.

All 108 identified patients identified in NCRAS and not originally identified by the screening units have subsequently had a confirmed diagnosis of recurrence through further checking with the hospital of recurrence, as identified via NCRAS. In the UK the patient would not necessarily have their recurrence diagnosed and treated at the same hospital as the original diagnosis.

The further validation of Sloane data on 100 individuals recruited from University Hospitals of Derby and Burton NHS Foundation Trust (UHDB) yielded good data concordance, but some estimation of the statistical confidence in this estimate would be helpful.

Unfortunately, we are not able to provide an estimate of statistical confidence for this as we do not have validation sets from other individual centres to compare this centre with. At present we are able to say that between 92% and 95% of further events are likely to be matched correctly by our methodology. However, further validation would be valuable going forward.

In the event definition table: What does contralateral re/occurrence mean? If contralateral recurrence is included in this term, it implies that a prior contralateral DCIS or invasive cancer had been diagnosed. But from the inclusion criteria it would appear that someone with prior contralateral disease would've been excluded. So in fact, according to the cohort entry criteria, only a new contralateral event would be relevant.

We apologise and thank the referee for pointing this out, and we have updated Supplemental Table 1, removing the 2 contralateral event re/occurrence definitions and adding a contralateral event definition.

In the same table, circulatory non-circulatory deaths are distinguished. What is the purpose of this distinction?

Because of the potential for radiotherapy-induced circulatory events we consider it to be potentially useful to be able to have these as a separate category. A note explaining this has been added to the legend for this table.

For non-UK practitioners, some terms will need explanation. For instance, it is not clear what is meant by a diagnostic versus a therapeutic specimen radiograph.

An alteration has been made to Table 2 to describe Specimen X-Ray. The Sloane radiology and pathology protocols which describe some of the terms are included in the references, and we have added these in the supplementary materials as well.

VERSION 2 – REVIEW

REVIEWER	Seema A. Khan Northwestern University
REVIEW RETURNED	02-Nov-2022
GENERAL COMMENTS	no further revisions